# The splenic T cell receptor repertoire during an immune response against a complex antigen: Expanding private clones accumulate in the high and low copy number region

**Martin Meinhardt**[☉], **Cornelia Tune**[☉], **Lisa-Kristin Schierloh, Andrea Schampel, René Pagel, Jürgen Westermann**[ID] *

Institute of Anatomy, University of Lübeck, Lübeck, Germany

[☉] These authors contributed equally to this work.

* juergen.westermann@uni-luebeck.de

**Data Availability Statement:** All relevant data are within the manuscript and its Supporting information files.

## Abstract

Large cellular antigens comprise a variety of different epitopes leading to a T cell response of extreme diversity. Therefore, tracking such a response by next generation sequencing of the T cell receptor (TCR) in order to identify common TCR properties among the expanding T cells represents an enormous challenge. In the present study we adapted a set of established indices to elucidate alterations in the TCR repertoire regarding sequence similarities between TCRs including VJ segment usage and diversity of nucleotide coding of a single TCR. We combined the usage of these indices with a new systematic splitting strategy regarding the copy number of the extracted clones to divide the repertoire into multiple fractions for separate analysis. We implemented this new analytic approach using the splenic TCR repertoire following immunization with sheep red blood cells (SRBC) in mice. As expected, early after immunization presumably antigen-specific clones accumulated in high copy number fractions, but at later time points similar accumulation of specific clones occurred within the repertoire fractions of lowest copy number. For both repertoire regions immunized animals could reliably be distinguished from control in a classification approach, demonstrating the robustness of the two effects at the individual level. The direction in which the indices shifted after immunization revealed that for both the early and the late effect alterations in repertoire parameters were caused by antigen-specific private clones displacing non-specific public clones. Taken together, tracking antigen-specific clones by their displacement of average TCR repertoire characteristics in standardized repertoire fractions ensures that our analytical approach is fairly independent from the antigen in question and thus allows the in-depth characterization of a variety of immune responses.

## Introduction

T cells are key players of the adaptive immune system. They are involved in direct combating of pathogens as well as in the coordination of other parts of the immune system including the

**Funding:** JW recieved funding from the "Deutsche Forschungsgemeinschaft", DFG Grant SFB 654, C4 https://www.dfg.de/ The funders had no role in study design, data collection and analysis, decision to publish, or preparation of the manuscript.

**Competing interests:** The authors have declared that no competing interests exist.

B cell response [1, 2]. Despite the different functions of the T cell subpopulations, the T cell receptor (TCR) is the common tool which enables them to recognize peptides presented by other cells. In humans and mice ~95% of circulating T cells express the TCR variant that consists of a dipeptide of α—and β -chain [3, 4]. The immense variety of pathogens threating the organism induces the need of a receptor repertoire of high diversity, which is achieved by a genetic mechanism called somatic recombination. This process includes V(D)J segment recombination as well as random nucleotide insertion and depletion at the junction sites. Each chain contains three hypervariable areas termed complementary determining region (CDR1–3). The CDR3 region spans the junction sites of V(D)J segments and is the site of highest variability [3]. In addition, it is the prime site of antigen-interaction and thus the main determinant of specificity of each T cell receptor/clone.

In mice, the number of different TCRs which can theoretically be generated is estimated as ~$10^{15}$ [5, 6], from which only ~$2 \cdot 10^6$ are realized at a given time point [5]. T cells that display CDR3 sequences commonly found in many individuals are termed "public" clones [7], while so-called "private" clones hold CDR3 sequences realized only in a few individuals. In case of antigen exposure, the number of reacting T cells is—even in the case of huge cellular antigens like bacteria – extremely low compared to the total number of T cells (between 0.01% and 0.1%, [8]) which renders their detection within the T cell receptor repertoire (TCR-R) determined by next generation sequencing a challenging bio-statistical task. A common strategy is to focus on T cell clones with highest copy number (CN) assuming that reacting and thus expanding clones exceed the CN of non-reacting ones (see e.g. [9–11]). For example, 3 days after immunization with SRBC a clear effect was observed among the splenic high CN (top 100) clones that quickly vanished within one day [12, 13], although T cell proliferation in this large antigen model continues well beyond this time point [13, 14]. This demonstrates the need for additional analytical strategies that can detect the effects induced by ongoing proliferation of presumably highly diverse T cell clones within the TCR-R.

In addition, even after expansion less frequent reacting clones might not reach CN levels higher than that of more frequent non-reacting ones. In line with this hypothesis, a classification approach after immunization with ovalbumin failed to discriminate between immunized and control animals based on clones with highest CN but was successful using clones detected with a single copy only [10]. Therefore, the aim of the present study was to improve the in-depth analysis of an immune response against a huge cellular antigen at the clonal level, for which we used the above-mentioned splenic TCR-R after immunization with SRBC [12–15]. We applied and adapted established indices which capture repertoire characteristics between and within animals exploiting both amino acid sequence and nucleotide coding level. In combination therewith we introduced a systematic fractioning strategy of the TCR-R that allowed the identification of those CN fractions in which antigen-specific clones accumulate. Our results showed that the analytical approach presented here significantly improved the in-depth analysis of TCR-Rs harboring highly divers immune reactions. We identified two separate regions of the splenic TCR-R where specific private clones accumulate at different time points following immunization with SRBC.

## Results

### Specific adaptions of the Simpson index reveal diversification of the TCR-R accompanied by decreasing diversity of nucleotide sequences which code for a given clonotype, for up to 7 days after immunization

To analyze the TCR-R during an immune response in the spleen we split the repertoire into three fractions of clonotypes according to their CN. Throughout this study the term

'clonotypes' refers to a set of T-cell clones with equal CDR3 amino acid sequence. To each of these clone sets exactly one V- and J-segment was assigned (see Material and method):

$CN^{low}$: CN = 2
$CN^{med}$: 2 < CN ≤ 500
$CN^{high}$: CN > 500

This tripartition ensures that the $CN^{high}$ fraction contains on average ~100 clonotypes, so this fraction is comparable to the top 100 clonotypes/clones of highest copy number which are analyzed separately in many studies [10, 12, 15]. Since we excluded sequences with CN = 1 from our analyses (see Material and method section), CN = 2 represents the group of lowest CN possible. We compared PBS-injected control mice to SRBC-immunized mice 3 days (3d), 4d and 7d after immunization and found an increase in clonotype number and mean CDR3β sequence length, which was restricted to both the 3d time point and the $CN^{high}$ fraction (Fig 1A and 1B). To assess possible expansion-induced shifts within both more complex and more sensitive repertoire parameters we converted the Simpson Index into a generalized version (see Material and methods for mathematical procedure) that quantifies the appearance of similar clonotypes within the repertoire of an animal and was therefore termed *Repertoire Homogeneity Index* (RHI). The advantage of the adapted version is that similarity can be defined within a variety of parameters depending on the research question. We here used the Levenshtein distance to capture the homogeneity of CDR3 sequences ($RHI_{LD}$; two clonotypes are defined as similar if the Levenshtein distance of their CDR3β regions is at maximum 1, note that a Levenshtein distance of zero implies equality of the two clonotypes). To capture the homogeneity of gene usage we considered the V- and J-segments which were assigned to the clonotypes ($RHI_{VJ}$; two clonotypes are defined as similar if equal V- and J-segments were assigned). Thus, for both $RHI_{LD}$ and $RHI_{VJ}$ an increase in value reflects an homogenization of the TCR-R.

Due to degeneration of the genetic code each clonotype can be encoded by different nucleotide sequences (i.e. consist of several actual T cell clones). To elucidate if immunization with SRBC also induces a shift at this level of the repertoire we used an alternative adaption of the Simpson Index to measure how diverse the nucleotide coding of each clonotype is (see Material and methods for mathematical details). We incorporated not only the number of nucleotides coding for the respective clonotype but also their proportion of sequence reads in such that a value close to 1 indicates rather balanced coding by multiple nucleotide sequences and a value close to 0 reflects predominant coding by a single nucleotide sequence. Accordingly, we named the subsequently calculated average of all clonotypes per animal *Coding Diversity Index* (CDI) for which an increase in value indicates a diversification of clonotype coding.

The significant decrease of $RHI_{LD}$ at 3d after immunization in the $CN^{high}$ fraction (Fig 1C) showed that the increase of clonotype number at this time point (Fig 1A) is associated with a diversification of immunized repertoires compared to control. Application of the $RHI_{VJ}$ (Fig 1D) revealed a diversification of gene usage within the $CN^{high}$ fraction that lasted until 4d and that a second diversification effect became clearly visible among the $CN^{low}$ fraction 7d after immunization. In parallel, the CDI shows that these timely separated diversification effects within the $CN^{high}$ and $CN^{low}$ fractions of the TCR-R were accompanied by a decreasing diversity of clonotype coding (Fig 1E), i.e. most of the expanding clonotypes are coded by few dominant nucleotide sequences.

Taken together, basic parameters such as CN and CDR3β sequence length identified immunization effects within the TCR-R only 3d after immunization with SRBC and only within the $CN^{high}$ fraction [12, 13, 15]. However, applying the newly created $RHI_{LD}$, $RHI_{VJ}$ and CDI made it possible to detect immune response-induced alterations within the TCR-R also among clonotypes of lowest CN and to trace these effects throughout all time points. Both effects

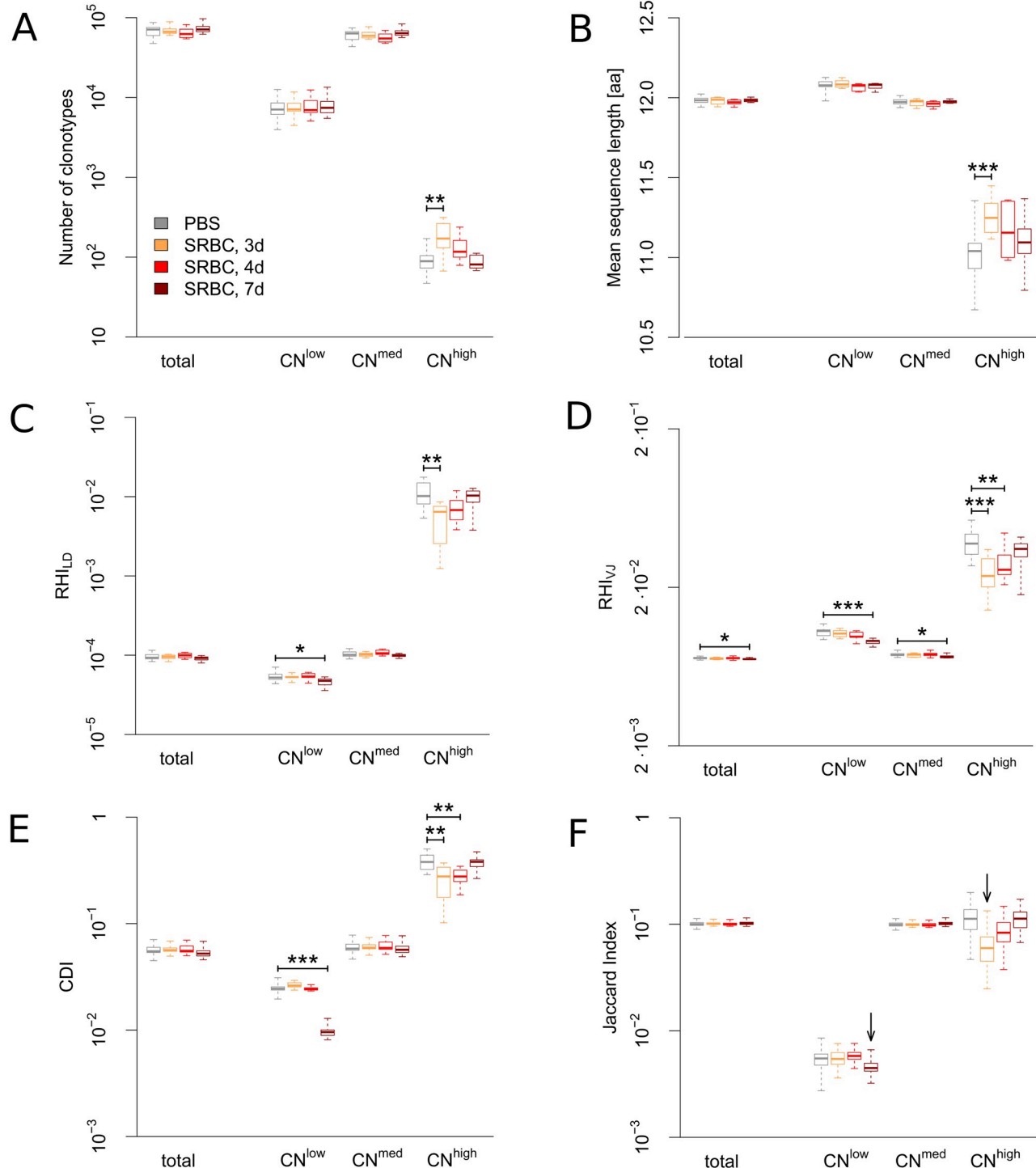

**Fig 1. Immunization with SRBC induces consistent parameter shifts in two distinct copy number fractions of the T cell receptor repertoire.** The data are presented either as total T cell receptor repertoire (containing all clonotypes) or divided into three clonotype fractions according to their copy number: clonotypes of low (CN = 2), intermediate ($2 < CN \leq 500$) and high (CN > 500) copy number. The total T cell receptor repertoire and each copy number fraction were compared 3, 4 and 7 days (d) after injection of SRBC (n = 10 per time point) with control (PBS-injected) animals (n = 20). (A) Number of clonotypes, (B) Mean sequence length of the CDR3β region, (C) Repertoire Homogeneity Index (RHI) capturing the homogeneity of CDR3β sequences within animals (two clonotypes defined as similar if the Levenshtein distance of the CDR3β regions is at maximum one), (D) RHI capturing homogeneity of gene usage (two clonotypes defined as similar if equal V- and J-segments were assigned), (E) Coding Diversity Index (CDI) capturing the diversity of nucleotide coding of the CDR3β amino acid sequence of each clonotype, and (F) Jaccard Index measuring the clonal overlap of CDR3β sequences from different animals. Boxplots display median, interquartile range and minima/maxima. For (A)-(E) immunized repertoires

were tested for deviations from control repertoires using Mann-Whitney-U-test. *p*-values are displayed as * $p < 0.05$, ** $p < 0.01$, *** $p < 0.001$. Correction for multiple testing was performed using Holm's method. For (F) apparent immunization effects are highlighted by arrows (see Material and methods).

showed diversification of repertoires (indicated by a decrease of RHI) that was accompanied by decreasing diversity of clonotype coding (indicated by a decrease of CDI). To elucidate if these alterations within animals also lead to changes between animals and thus repertoire shifts at the population level we applied the Jaccard Index and found it reduced both at 3d after immunization within the $CN^{high}$ fraction and at 7d within the $CN^{low}$ fraction (Fig 1F) showing that in each animal different clonotypes reacted to the SRBC antigens which indicated the private nature of this immune response [7, 12, 15].

## Systematic fractioning of the TCR-R by clonotype copy number reveals the full extent of two clearly separated repertoire regions affected by immunization

The significant decrease of $RHI_{VJ}$ within the $CN^{med}$ fraction 7d after immunization (Fig 1D) raised the question how far the effect detected by all indices in the $CN^{low}$ fraction extended from the bottom into the $CN^{med}$ fraction. Similarly, the early effects within the $CN^{high}$ fraction might extend into the $CN^{med}$ fraction from the top due to huge differences in the number of clonotypes in the two fractions (on average about 100 within $CN^{high}$ and about 60, 000 within $CN^{med}$; Fig 1A). Therefore, we aimed to refine the previously arbitrary fractioning by further and systematical splitting of the intermediate fraction in such that: i) the number of fractions is kept as low as possible to avoid the loss of true effects due to correction for multiple comparisons but high enough for a precise mapping of immunization effects to distinct fractions, and ii) the number of clonotypes in each fraction is high enough to yield statistically valid results. While these criteria ruled out a linear approach, we found that fractioning the total repertoire by CN based on the logarithm to the base 2 fulfilled these requirements: resulting in 10 fractions with the first fraction with $\log_2(CN) = 1$ (i.e. CN = 2) equaling the $CN^{low}$ fraction and the last with $\log_2(CN) > 9$ (i.e. CN > 512) corresponding to the $CN^{high}$ fraction, while the ~60, 000 clonotypes of the intermediate fraction now distributed over 8 fractions. The number of clonotypes per fraction in a single data set ranged from a maximum of about 21, 000 to a minimum of 45 (Fig 2A). While basic parameters such as number of clonotypes (Fig 2A) and mean sequence length (S1A Fig) remained unaltered in all but the fraction of highest CN 3d after immunization (the mean sequence length also at 4d within the second highest copy fraction), the refined fractioning already payed off concerning the Jaccard Index, where in addition to the known decreases in $CN^{high}$ at 3d and $CN^{low}$ at 7d obvious decreases appeared in the two fractions from $2^6$ to $2^9$ at both 3d and 4d after immunization (S1B Fig).

Subsequently, we compared $RHI_{LD}$, $RHI_{VJ}$ and CDI of the immunized animals for each time point and fraction to that of control animals, visualizing the resulting *p*-values in a heat map for easy comparison (Fig 2B). Our results showed that fractioning the total repertoire by CN based on the logarithm to the base 2 paired with application of the adapted indices allowed an in-depth analysis of the splenic TCR-R during a SRBC-induced immune response that revealed three main findings (Fig 2B): First, the V- and J-segment usage measured by $RHI_{VJ}$ harbored the greatest discriminatory power by detecting immunization effects among clonotypes of high and medium CN until 7d after immunization as well as among clonotypes of low CN already 4d after immunization, while the CDI displayed a superior discriminatory power for the immunization effect within the two fractions of lowest CN 7d after immunization.

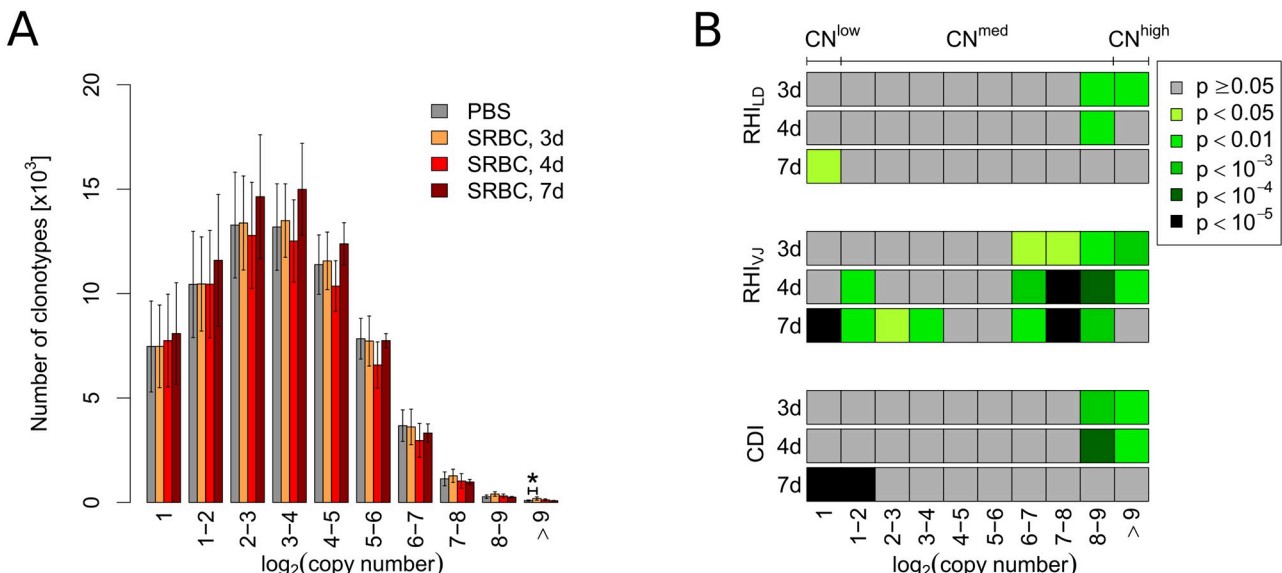

**Fig 2. Logarithmic partitioning leads to repertoire fractions of appropriate size for separate analyses that reveal true extent and separation of two immunization-induced effects within the repertoire.** Total repertoires where split into 10 fractions according to their copy number based on the logarithm to base 2. (A) Number of extracted clonotypes per fraction. Bars and whiskers display means and standard deviations. Repertoires of animals 3, 4 and 7 days (d) after immunization with SRBC (n = 10 each) were tested for deviations from the control (PBS-injected) repertoires (n = 20) using the Mann-Whitney-U-test. $p$-values are displayed as $^*$ $p < 0.05$. Correction for multiple testing was performed using Holm's method. (B) For each fraction and time point immunized samples were compared to control samples for three parameters: the Repertoire Homogeneity Index (RHI) assessing repertoire homogeneity concerning either i) CDR3β sequence similarity measured by the Levenshtein distance ($RHI_{LD}$) or ii) VJ segment usage ($RHI_{VJ}$) as well as iii) the Coding Diversity Index (CDI) assessing heterogeneity of clonotype coding. $p$-values refer to a one tailed Mann-Whitney-U-test with significant decrease of the respective index. The 10 $p$-values of each row were independently corrected for multiple testing using Holm's method.

Second, 7d after immunization none of the indices reached significance within the fraction of highest CN, while the immunization effect was still prominent several fractions below ($6 < \log_2(CN) \leq 9$) at least for $RHI_{VJ}$. This remained true even if the $p$-values were not corrected for multiple testing demonstrating that significant effects were not hidden by statistical correction steps (S1C Fig). Third, the early effect among clonotypes of medium to high CN and the late effect within fractions of lowest CN were separated by a clear gap of at least two fractions ($4 < \log_2(CN) \leq 6$, about 19, 000 clones total and thus nearly a quarter of the whole repertoire) that did not display alterations in any of the repertoire parameters, even without correction for multiple testing (S1C Fig).

Thus, systematic fractioning of the splenic TCR-R by CN clearly revealed two discrete sites of diversification on the animal level, one occurring early during the immune response within the high CN region and one occurring late within the low CN region, both being accompanied by a homogenization of nucleotide coding on the clonotype level.

## Classification becomes successful when performed separately on the high and low copy number sub-repertoires that display immunization-induced effects

Our results revealed a variety of significant alterations of repertoire characteristics ranging from the animal level (clonotype similarities measured by the RHI) to the clonotype level (homogenization of nucleotide coding measured by the CDI) when the group of control animals was compared to that of immunized animals. To investigate whether these effects were pronounced enough to be demonstrated within individual animals, we constructed a simple

classification tool. In brief, we used a generalized version of Morisita-Horn Index [16, 17] to define a measure of dissimilarity between to TCR-Rs. Subsequently the data sets were arranged in clusters via K-medoid clustering [18, 19]. A slight modification leads to a supervised classification procedure (see Material and methods). We analyzed, if this algorithm can reliable distinguish TCR-Rs of immunized animals from those of naïve. We hypothesized that focusing on those repertoire regions in which reacting clones accumulate will lead to a successful classification. For definition of the two relevant sub-repertoires, we used those fractions that reached significance either at least at two different time points or for two of the three indices, leading to

$X^{\text{top}}$: $\log_2(\text{CN}) > 6$, i.e. CN > 64 and

$X^{\text{bottom}}$: $\log_2(\text{CN}) \leq 2$, i.e. CN $\leq$ 4.

Considering VJ segment usage as criterion, unsupervised classification of $X^{\text{top}}$ arranged the sub-repertoire of high CN in a cluster structure where one cluster is clearly dominated by control (PBS) and the other by immunized samples (SRBC) indicated by triangles and circles, respectively (Fig 3A). While most overlap between the clusters was caused by mice of the control and 3d SRBC group, all but one animal each of the 4d and 7d group were assigned into the

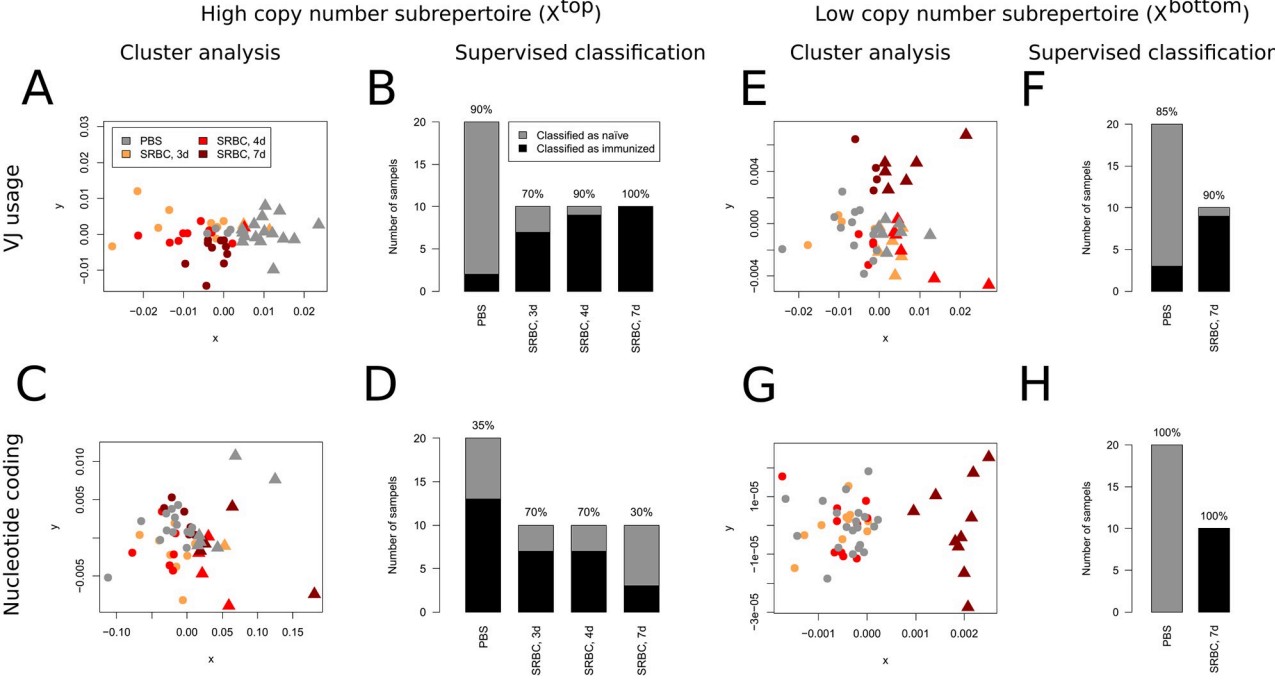

**Fig 3. Sub-repertoires can reliable be classified in view of the immunization status depending on the classification criterion.** (A)-(D) The high copy number sub-repertoire $X^{\text{top}}$ was defined as clonotypes with copy number >64. (A) A dissimilarity matrix was calculated based on VJ segment usage (see Material and methods) by which these sub-repertoires were divided into two clusters (immunized vs. control) using the K-medoid algorithm and the result visualized via metric multidimensional scaling with dissimilarities approximated by distances of points in the scatterplot and clusters defined by the algorithm displayed as circles and triangles, respectively. (B) The same dissimilarity matrix was used for a supervised classification that in total assigned 44 of the 50 data sets correctly. The number on top of each bar denotes the percentage of correctly classified samples for each of the 4 experimental groups. (C)-(D) The $X^{\text{top}}$ sub-repertoire data were classified based on the variability of clonotype nucleotide coding, which fails to distinguish any of the experimental groups from the others. In total, 22 of the 50 data sets were classified correctly which does not significantly outstrip random labeling ($p = 0.377$). (E)-(H) The low copy number sub-repertoire $X^{\text{bottom}}$ was defined as clonotypes with copy number $\leq$ 4 and subjected to the same classification approaches as the $X^{\text{top}}$ sub-repertoires, with the exception that for supervised classification the control data sets were compared to immunized samples of the SRBC group 7 days (d) after immunization only. Although the VJ usage as criterion did not result in obvious clustering (E) the supervised approach (F) significantly outstripped random classification ($p < 2 \cdot 10^{-4}$). The nucleotide coding criterion led to a well separated cluster formed by the 7d SRBC group (G) that could be classified with perfect accuracy (H). In contrast, the $X^{\text{bottom}}$ sub-repertoire of animals 3d and 4d after immunization were indistinguishable from corresponding sub-repertoires of control animals.

same cluster. This was confirmed by the supervised approach that classified 18 out of 20 control mice and 26 out of 30 immunized mice correctly (Fig 3B), which differed significantly from random labelling ($p < 6 \cdot 10^{-8}$). However, applying the nucleotide coding criterion to the $X^{\text{top}}$ sub-repertoire neither resulted in obvious clustering (Fig 3C) nor significantly outstripped random classification (Fig 3D, $p = 0.377$). In contrast, when applying cluster analysis to the bottom sub-repertoire, VJ segment usage as criterion failed to distinguish between the experimental groups, with only the 7d group slightly separating from the remaining samples (Fig 3E). For supervised classification we even had to remove the 3d and 4d samples to avoid disrupting the algorithm, since at this time point only marginally immunization effects were detectable in the respective fractions (Fig 2B). Subsequently, supervised classification distinguished animals of the 7d group from control samples with moderate accuracy (Fig 3F, $p < 2 \cdot 10^{-4}$). Analog to the highly significant differences in CDI (see Fig 2B), the nucleotide coding criterion for $X^{\text{bottom}}$ on the other hand led to one cluster that exactly coincides with the 7d group, whereas the other cluster contained the remaining data sets (Fig 3G). Correspondingly, the supervised approach led to a perfect classification (Fig 3H; random effects: $p < 4 \cdot 10^{-8}$). It should be mentioned that a distinct improvement of the classification based on nucleotide coding can be achieved, if the definition of $X^{\text{top}}$ is modified in this way that only clonotypes with $\log_2(CN) > 8$ are included (i.e. the fractions were CDI is affected significantly). In particular the 3d and 4d samples can be distinguished from naïve with satisfying accuracy (data not shown). When total repertoires were analyzed without weighting for copy number both cluster analysis and supervised classification approaches based on VJ segment usage failed to distinguish any of the immunized groups from naïve (S2A and S2B Fig). If the nucleotide criterion is applied, only the samples of the 7d group are slightly separated from the remaining samples, displaying a similar pattern as for $X^{\text{bottom}}$. This can easily be explained by the large quantity of clonotypes in these fractions. In particular the 3d and 4d data were indistinguishable from the naïve (data not shown). Thus, focusing on those repertoire regions in which the immune response was localized using the fractioning approach allowed for classification of individual TCR-R with satisfying accuracy.

## Expanding public clonotypes distribute throughout the repertoire fractions, but accumulate in the fraction of highest copy number after immunization with SRBC

The SRBC-specific effects described so far were mainly caused by an immune response involving private clonotypes as demonstrated previously [12, 15] and in the present study by a decrease of the Jaccard index (Fig 1F and S1B Fig). However, in these previous studies we were able to also identify a small set of clonotypes that was present in the majority of SRBC-immunized animals and significantly expanded compared to control [12, 13]. The expanding public clonotypes were identified via differential gene expression analysis using the R-package *edgeR* [20] (see Material and methods). Although the specificity of the expanding clonotypes was not tested explicitly, the observed enrichment can be seen as a hint that they react to an epitope derived from the injected SRBC, thus they can be seen as a public component of the SRBC induced T-cell response. Here, we now ask how these clonotypes distribute within the different CN fractions of the TCR-R over time.

Within the control group presumably SRBC-specific clonotypes displayed a bell-shaped distribution over the ten repertoire fractions defined above. Thereby the highest number of clonotypes was found in the $5 < \log_2(CN) \leq 6$ fraction (Fig 4). In contrast to this, considering the total distribution of all clonotypes, the highest number of clonotypes was found in the $2 < \log_2(CN) \leq 4$ fractions (see Fig 2A). After the immunization with SRBC the expansion

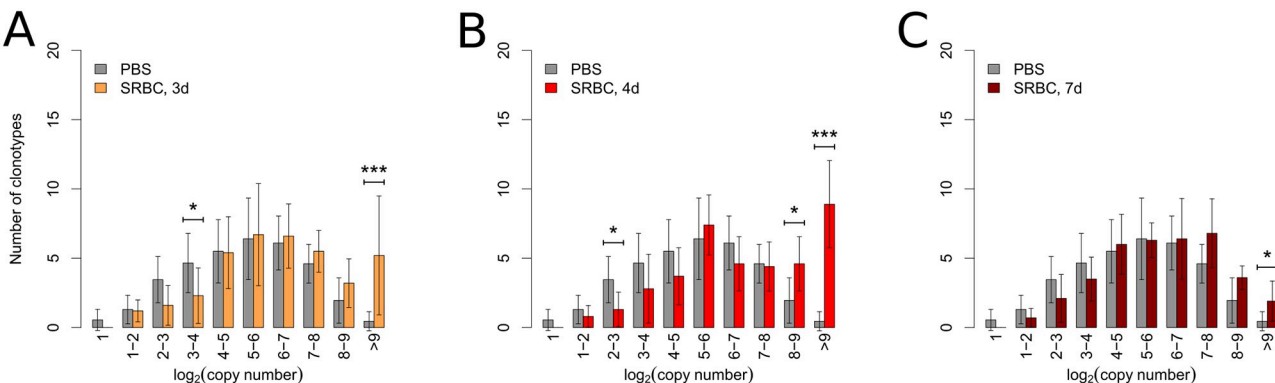

**Fig 4. Expanding public clonotypes distribute throughout the repertoire and accumulate in the fraction of highest copy number after immunization.** 40 clonotypes detected in the majority of immunized animals and found significantly expanded after SRBC injection were defined as (presumably) SRBC-specific public clonotypes. Their distribution throughout the 10 fractions defined by copy number based on the logarithm of the base 2 in PBS-injected control animals is indicated by grey bars compared to animals 3 days (d) (A), 4d (B) and 7d (C) after immunization with SRBC. Barplots state number of SRBC-specific clonotypes per fraction for control (n = 20) and immunized (n = 10 for each time point) mice. Bars and whiskers display means and standard deviations. For each time point control and immunized animals were compared using the Mann-Whitney-U-test with resulting $p$-values displayed as $^{*}$ $p < 0.05$, $^{***}$ $p < 0.001$. $p$-values were corrected for multiple testing using Holm's method.

effects led to an accumulation of presumably specific public clonotypes within the high CN fraction. This prominent effect persisted until 7d p.i. (Fig 4C). Thus, the immunization effects among public and private clones (the latter are summarized in Fig 2B and S1C Fig) match at 3d and 4d with similar accumulation of clonotype numbers in the high copy fraction. In difference to this, the immunization effect of the private component failed to reach significance within the high CN fraction 7d p.i (see S1C Fig). Apart from the accumulation in the high CN fraction the immunization resulted in significant differences between control group and immunized mice in a few separate fractions at 3d and 4d. At all time points more than a quarter of the expanding public clonotypes were located in the $4 < \log_2(\text{CN}) \leq 6$ fractions where no significant 'private activity' was detected at any time points investigated (see S1C Fig). In contrast to the predominant private component of the reaction, there is no accumulation of expanding public clonotypes in the low CN fraction. This effect can be explained by the fact that public clonotypes often descend from various progenitor cells whose descendants can not be distinguished in our experimental setting (see discussion).

## Discussion

### Tracing highly diverse antigen-specific T cells during an immune response at the TCR-R level

The aim of this study was to analyze an immune response where the antigen-specific TCR-R is too heterogeneous to be identified by common approaches searching for e.g. clusters of CDR3 similarity. Via specific adaptions of established indices we monitored immunization-induced shifts of repertoire characteristics within different aspects of the TCR-R, first by measuring repertoire homogeneity concerning CDR3β sequence similarity ($\text{RHI}_{LD}$) and VJ segment usage ($\text{RHI}_{VJ}$) and second by depicting nucleotide coding diversity (CDI). While the sensitivity of $\text{RHI}_{VJ}$ was potent enough to detect the presumably minuscule shifts within the whole repertoire, an arbitrary separation into three repertoire fractions clearly revealed shifts for $\text{RHI}_{VJ}$, $\text{RHI}_{LD}$ and CDI within both the low and high CN fractions. However, only systematic fractioning of the TCR-R by CN based on the logarithm to the base 2 resulting in 10 fractions revealed the full extend and detailed time course of the two immune response-induced

parameter shifts, which allowed us to demonstrate two important aspects: First, early immunization induced effects at 3d and 4d could be located not only among clonotypes of highest CN but also within the upper four CN fractions (on average matching the top 5,000 clonotypes). Furthermore, 7d after immunization the same pattern was presented with the important exception that the highest CN fraction was not affected anymore (Fig 2B). This might explain why studies analyzing only clonotypes with highest CN at late time points after immunization failed to demonstrate immune response-induced effects on the TCR-R [10, 12, 13]. Second, our approach revealed a late immunization-induced shift within the fractions of lowest CN manifesting 7d after immunization and thus considerably later than the first effect. This effect was seen in all parameters used and confirmed by significant classification results and has to our knowledge not been characterized before. Throughout time points and parameters, the two effects observed within the fractions of high and low CN were separated by at least two fractions of medium CN that did not show any immunization effects indicating that the observed effects are caused by two different biological mechanisms.

## Expansion and migration of SRBC-specific clonotypes cause the effects observed in the high and low copy number fractions

The final expansion level of a clonotype is assumed to be determined by receptor affinity, antigen dose and competitive pressure of other clonotypes [21, 22]. Thus, clonotypes which first reach the site of antigen presentation (here: the spleen) expand under conditions of overabundance of antigen and without any competitive pressure, leading to a massive expansion that manifests in an increased number of clonotypes in the top CN fractions. Subsequently, since SRBC are a non-replicating antigen, over time an increasing number of specific clonotypes compete for a decreasing amount of antigen which leads to a continuous decrease in expansion rates. This explains the accumulation of reacting clones in fractions of medium CN and the lack of immunization effects in the fraction with highest CN at later time points. Furthermore, both the egress of SRBC-specific clonotypes and simultaneous entry of non-specific clonotypes contribute to the descent of the immunization-induced parameter shifts to fractions of medium CN. The second effect of the T cell response that manifested among clonotypes of lowest CN is probably due to the (re-)immigration of antigen-specific T cells that originally expanded in other parts of the spleen. The observation that the number of SRBC-specific clonotypes in the blood increases 4d after immunization strongly supports this notion [13]. Further experiments are needed to determine whether characteristics and time course of the described alterations of the TCR-R are typical for immune responses against large, non-replicating antigens or if they also hold true for small and/or replicating antigens.

## The immunization effects revealed by RHI and CDI within the TCR-R are due to the displacement of unspecific public clonotypes by specific private clonotypes

The reduction of the Jaccard Index measuring the clonal overlap between animals already confirmed the predominantly private nature of both immunization-induced effects (Fig 1F). Thus, expanding SRBC-specific private clonotypes displace naïve clonotypes within both affected regions of the repertoire. The applied indices do not detect the displacement of the many unspecific private clonotypes but instead that of the few unspecific public clonotypes which are known to appear in considerably higher CN than private clonotypes [23]. This is due to their generation by a combination of increased probability of occurrence during somatic recombination [24, 25] and selective advantages during positive and negative selection [23], which also explains their on average shorter CDR3β length. These features are reflected by

both the decrease of mean sequence length and increase of the Jaccard index in the $CN^{high}$ fraction (Fig 1B and 1F) and throughout the ten repertoire fractions (S1A and S1B Fig), respectively. Furthermore, the three parameters recognized by $RHI_{LD}$, $RHI_{VJ}$ and CDI and their change of direction fit well into the interpretation that the immunization-induced effects emerge by specific private clonotypes displacing non-specific public clonotypes: First, public clonotypes are arranged in cluster structures where the public prototype is 'surrounded' by a set of (not necessarily public) clonotypes with very similar CDR3β regions [26]. Thus, the reduced number of similar CDR3β sequences as quantified by a decrease of $RHI_{LD}$ can be ascribed to the disappearance of the public cluster structures due to the expansion of SRBC-specific private clonotypes (Fig 1C). Second, public clonotypes display a restricted VJ segment usage [23] which leads to relative homogeneity of VJ segment usage within the different CN fractions. After immunization with SRBC the affected CN fractions are less homogenous as indicated by the $RHI_{VJ}$ index (Fig 1D) because public clonotypes are displaced by SRBC-specific private clonotypes. The superior discriminatory power of $RHI_{VJ}$ compared to $RHI_{LD}$ is due to the biological features of the immune response against SRBC and could as well be the other way round in responses against other antigens when dominated by clonotypes that rather share similar CDR3β sequences than VJ segment usage. Third, convergent recombination leads to the appearance of public clonotypes in families with identical CDR3β amino acid sequences (i.e. one clonotype) which are encoded by several nucleotide sequences [23]. This is only rarely the case for private clonotypes. Therefore, a decrease of nucleotide coding diversity as measured by the CDI indicates the displacement of public clonotypes due to an increase in the number of SRBC-specific private clonotypes within the respective CN fractions (Fig 1E). Taken together, the decrease in homogeneity regarding CDR3β amino acid sequence ($RHI_{LD}$) and VJ segment usage ($RHI_{VJ}$) in combination with the decrease in nucleotide coding diversity (CDI) demonstrates the displacement of public clonotypes by SRBC-specific private ones. Most likely, however, RHI and CDI would be also able to monitor immune responses at the TCR-R level that predominately are of public nature. Here the expansion of specific public clonotypes would lead to repertoire shifts in $RHI_{LD}$, $RHI_{VJ}$ and CDI in the opposite direction as found in the present study for SRBC due to the replacement of unspecific private clonotypes. In addition, the extremely small public component among a SRBC-induced immune response can only be followed by directly tracing a set of previously identified SRBC-specific public clonotypes [12, 13, 15]. In contrast to the private component, an increase of SRBC-specific public clonotypes was observed mainly in the high CN fraction and lacking in the fraction of lowest CN (Fig 4). This observation can be explained by limitations of our analyzing approach. Activated T cells with public CDR3β sequence which remigrate from the blood into the spleen usually meet local proliferating 'conspecifics' i.e. T cells of the same clonotype. In our setting remigrating and local expanding T cells of the same clonotype can not be distinguished. Thus, the extracted CDR3β sequences are merged and ascribed to a fraction of higher CN.

## Classification approaches highlight the importance of systematic repertoire fractioning for in-depth analysis of the TCR-R

Detection of immunization-induced effects in this model failed when either the whole repertoire was analyzed or the CN fractions were defined too broad (Fig 1). In addition, also classification was impossible when whole repertoires were investigated (S2A and S2B Fig). Interestingly, the latter could partly be solved by weighting for CN (see S1 Methods and S2C and S2D Fig). However, such approaches are in principle unable to reveal the two separated immunization-induced effects. Especially the late effect among clonotypes of lowest CN and its clear separation from the early effect within high CN fractions could only be elucidated by the

refined fractioning of the whole repertoire following the systematic and objective strategy based on the logarithm to the base 2. This system leads to an optimal combination of clonotypes per fraction and number of fractions for subsequent statistical analysis that can be applied to any repertoire, allowing precise detection and discrimination of immunization-induced effects. Subsequently, separate classification approaches within those repertoire regions where antigen-specific clonotypes accumulate can reach satisfactory success rates as shown here for immunization with SRBC (Fig 3) confirming the occurrence of distinct accumulations of antigen-specific clonotypes at the level of individual animals. Thus, identification of CN fractions containing significant numbers of antigen-specific clonotypes as outlined in the present study might reveal important insights in the dynamics of the T cell response against a variety of antigens.

## Conclusion

Combining RHI and CDI and/or classification approaches using analog discriminatory parameters with systematical splitting of the TCR-R into different CN fractions allows to reveal the dynamics of the SRBC-specific T cell response at the TCR-R level. We demonstrate that SRBC-specific clonotypes first accumulate in high CN fractions and at later time points also in low CN fractions. The early expansion-based effect within high copy fractions extends also into the medium CN fractions together containing approximately the top 5,000 clonotypes. Thus, selective analyses of considerably smaller top fractions like the top 100 clonotypes as done in our previous [12, 15] or other studies investigating non-replicating antigens such as ovalbumin [10] might miss relevant information. The present study shows that although no alterations are observed in the highest CN fraction (comparable to the top 100) at 7d, clear shifts are seen within the three fractions below (Fig 2B). The late migration-based effect among clonotypes of lowest CN has not been described before. Thus, the analytic strategy outlined in the present study allows the precise localization and characterization of immunization effects at the TCR-R level.

## Materials and methods

The present work is based on data derived from a previously published study [12]. Relevant experimental aspects such as rearrangement of experimental groups and data preprocessing are briefly described below. For protocols of animal handling, spleen removal and cryo-sectioning as well as total RNA extraction, CDR3β-chain transcription and amplification we refer to the initial publication [12].

### Mouse model and experimental groups

Eight- to twelve-week-old C57BL/6J mice were either immunized by injecting 200 ml phosphate-buffered saline (PBS) containing $10^9$ SRBC into the tail vein or attributed as control receiving 200 ml PBS only. Immunized mice were sacrificed 3d, 4d and 7d after immunization (n = 10 each), PBS-injected animals at 3d and 4d (n = 10 each) and were merged into a single control group (n = 20). Originally, half of each group was experimentally exposed to short-term sleep restriction, but the T cell response was not affected by this manipulation [12], which is why we merged both conditions resulting in four groups: 'PBS' (control), 'SRBC 3d', 'SRBC 4d' and 'SRBC 7d' (days post-immunization, respectively).

### TCR repertoire generation

CDR3β identification, clonotype clustering and correction of sequencing errors were performed using MiTCR software [27]. All parameters were set to the standard values of the

ClonoCalc graphical user interface [28] for MiTCR. After removing nonfunctional sequences, we obtained an average of 1.9 million reads for each sample which were assigned to a mean of 100,000 different CDR3β nucleotide sequences. Due to experimental variability, the read counts in the 7d group were consistently increased (on average 2.7 million in the 7d SRBC group vs. 1.6 million in the remaining samples). To obtain comparable samples, the data sets of the 7d group were downsampled to the mean read count of the remaining data sets of 1.6 million reads. Note, that the downsampling of the 7d group was performed after removing nonfunctional nucleotide sequences. Thus, the final result of the downsampling step is not affected by these sequences. Subsequently, all nucleotide sequences coding for identical amino acid sequences were merged into one 'clonotype' that refers to a set of T cells with identical CDR3β region. Note that it is not ensured that all sequences of such clonotypes bear identical receptors since they can differ in subsequences outside the CDR3β region as well as in the α-chain. Subsequently, each clonotype was assigned the V- and J-segments of the underlying nucleotide sequence of highest read count as well as the summarized read count of all underlying nucleotide sequences as total read count, in the following referred to as copy number (CN). To avoid artificial clonotypes arising from polymerase reading errors during the sequencing procedure, sequences of copy number 1 were excluded from further analysis. After this, the number of clonotypes per repertoire ranged from approximately 50,000 to 100,000 per sample. The exact numbers for the extracted clonotypes and sequences of each sample are provided as (S1 Table).

**Standard parameters and statistics.**   For each animal we extracted number of clonotypes and mean CDR3β sequence length, either for the whole repertoire or certain fractions defined in the results section. Furthermore, we used the Jaccard Index to quantify the clonal overlap of the extracted TCR-R of different mice. We calculated this index for each potential pairing of data sets in each experimental group which leads to multiple dependencies between the obtained values. This violates basic assumptions of commonly used inferential statistics, e.g. calculation of $p$-values and confidence intervals, which is why we restricted our analysis on descriptive considerations for this parameter. For all standard parameters and other indices (see below), immunized and control repertoires were compared using the Mann-Whitney-U-test. Unless otherwise mentioned all tests were two-tailed with a limit of significance of 0.05. Correction for multiple testing was performed using Holm's method. $p$-values which refer to analyses of repertoire fractions were corrected independently from the analyses of total repertoires. Calculations and data visualization were performed using the R platform for statistical computing [29].

In the following, we refined established indices like the Simpson and Morisita-Horn Index in such that the resulting new indices allow for both a more flexible and deeper characterization of the TCR-R. A detailed description of underlying assumptions and derivations drawn is provided as (see S1 Methods), while only central aspects are presented in the following.

Each TCR-R data set $X$ can be interpreted as a finite set of pairs $(x_i, v_X(x_i))_{i = 1, \ldots, m}$, where $v_X(x)$ denotes the CN of a clonotype $x$ in $X$. To allow a comparison of clonotypes concerning complex parameters such as V- and J-segment usage and CDR3β sequence similarity we defined generalized versions of the indices mentioned above. In the following we denote by

- $\Omega$ the set of all possible TCRβ sequences,

- $R \subset \Omega \times \Omega$ an arbitrary reflexive, symmetric relation (i.e. a criterion of similarity of two sequences) on $\Omega$ and by

- $\mathbb{1}(\cdot)$ the indicator function (returning 1 if the given statement is true and 0 if not).

In dependence of the similarity criterion $R$ we defined the *Repertoire Homogeneity Index* as

$$\text{RHI}_R(X) = \frac{\displaystyle\sum_{\substack{i=1,\ldots,m \\ j<i}} \mathbb{1}(x_i R x_j)}{\displaystyle\binom{m}{2}}$$

Thereby RHI reflects the probability by which a randomly sampled pair of clonotypes within a data set is similar in respect of R, with R allowing a flexible definition of similarity within a variety of sequence parameters. The concept of this approach is analog to that of the Simpson Index [30, 31]. Note that the adapted index does not account for CN since we aimed to apply it on repertoire fractions already defined by CN (for a weighted version of the RHI, see S1 Methods). Subsequently, we defined as criterion of similarity R either that the two clonotypes share equal V- and J-segments ($\text{RHI}_{\text{VJ}}$), or that the Levenshtein distance (LD) [32] of their CDR3β regions is at maximum 1 ($\text{RHI}_{\text{LD}}$).

**Coding diversity index.** Each clonotype $x \in X$ can be encoded by several nucleotide sequences $x^{(1)}, \ldots, x^{(m_x)}$. Based on the Simpson Index we converted this nucleotide coding parameter into an index that accounts for the whole repertoire and thus allows comparison between animals and groups. For each clonotype $x \in X$ we first defined the Nucleotide Coding Simpson Index as

$$D_{\text{NC}}(x) = 1 - \sum_{i=1}^{m_x} \left( \frac{v_X(x^{(i)})}{v_X(x)} \right)^2.$$

$D_{\text{NC}}(x)$ quantifies the coding diversity of the clonotype $x$. The mean value of these indices provides a measure for the heterogeneity of the nucleotide coding of the total of the clonotypes in $X$. Accordingly, we defined this value as *Coding Diversity Index* which is given by

$$\text{CDI}(X) = \frac{1}{m} \sum_{i=1}^{m} D_{\text{NC}}(x_i).$$

**Cluster and classification analyses.** The construction of classification tools required a quantification of dissimilarity of two different data sets. In analogy of RHI capturing clonal similarities within a repertoire, we defined the *Repertoire Similarity Index* (RSI) that allows a comparison of clonotypes deriving from $X$ with clonotypes of another data set $Y = (y_i, v_Y(y_i))_{i=1,\ldots,n}$ and thus between repertoires as

$$\text{RSI}_R(X, Y) = \frac{2 \displaystyle\sum_{\substack{i=1,\ldots,m \\ j=1,\ldots,n}} \mathbb{1}(x_i R y_j)}{nm \left( \displaystyle\sum_{i,j=1,\ldots,m} \frac{\mathbb{1}(x_i R x_j)}{m^2} + \sum_{i,j=1,\ldots,n} \frac{\mathbb{1}(y_i R y_j)}{n^2} \right)}.$$

Here, in the special case of the similarity criterion $R$ demanding clonotype identity, $\text{RSI}_R$ coincides with the Sørensen Index [17] and the analog version weighting for CN (see S1 Methods) with the Morisita-Horn Index [16, 33]. Subsequently, for $k$ different relations $R_1, \ldots, R_k$ attributed with appropriate weights $\alpha = (\alpha_1, \ldots, \alpha_k)$ satisfying $\alpha_i \geq 0$, $i = 1, \ldots, k$ and

$\sum_{i=1}^{k} \alpha_i = 1$ the index

$$d_{\alpha, R_1, \cdots, R_k}(X, Y) = 1 - \sum_{i=1}^{k} \alpha_i \min(\text{RSI}_{R_i}(X, Y), 1)$$

provides the required measure of dissimilarity defined by the respective $k$ criteria. For the subsequent cluster and classification analysis of repertoire fractions defined by CN we considered two different versions of $d$. With the first we targeted VJ segment usage corresponding to $\text{RHI}_{\text{VJ}}$, with identity of V- and J-segment, respectively, treated as two independent and equally weighted criteria of similarity for calculation of $\text{RSI}_{\text{V}}$, $\text{RSI}_{\text{J}}$ and subsequently $d_{\text{V,J}}$. For the second we utilized the nucleotide coding (NC) of clonotypes by calculating $\text{RSI}_{\text{NC}}$ and $d_{\text{NC}}$ with clonotypes $x$ and $y$ considered similar if the number of different nucleotide sequences coding for the respective clonotypes either coincides or exceeds 5 for both sequences as single criterion. This corresponds to CDI but does not account for proportional read counts of the nucleotide sequences.

The data sets were categorized into two groups via K-medoids clustering [18, 19] using either $d_{\text{V,J}}$ or $d_{\text{NC}}$ as discriminatory criterion, with results visualized via metric multidimensional scaling [18] where dissimilarities are approximated by distances of points in a scatterplot. Supervised classification was performed using the leave-one-out procedure [34] with the training data sets labeled as 'control' and 'immunized'. Subsequently, the K-medoid algorithm was independently applied to each of the two groups defining exactly one medoid in each group. The label of the nearest medoid (prototype) was assigned to the only sample of the test data set. Note that classification criteria and subrepertoires were defined after analyzing the data sets and subsequently the same data were used as test data sets for the evaluation of the algorithm in view of discriminatory power. Such approaches might result in an overestimation of discriminatory power, in particular if sample specific properties of the data are incorporated in the algorithm, a problem discussed in detail in [35].

The results were evaluated using an exact Fisher test, with a significant result implying that the labels are indeed affected by the true immunization status (in contrast to random labeling of the samples). K-medoid clustering was performed using the *pam* function of the R-package *cluster* [36]. Calculation of RHI and RSI was performed in Java with the required software developed using Eclipse IDE for Java Developers. For calculation of the Levenshtein distance the Apache Commons Text library [37] was applied.

In general, the main advantage of RHI and RSI is a high degree of flexibility. In fact, arbitrary criteria for similarity between sequences (which are formalized as reflexive and symmetric relations) can be applied to quantify data sets in view of homogeneity and similarity. This ensures a universal applicability which is independent from the concrete biological parameters of interest. Their main disadvantage are the high computational costs in case of non-transitive relations where all potential pairs of clonotypes have to be evaluated in view of the similarity criterion leading to exorbitant computational time. In case of equivalence relations, the indices can be calculated using the original formula (Simpson or Morisita-Horn Index) and represent an application of well-established statistical methods. For example, Glanville *et. al* applied the Simpson Index to evaluate the homogeneity of V-segments and CDR3 sequence length in preselected TCR-clusters [38]. In addition to a dramatic reduction of computational time, the application of equivalence relations ensures some analytic properties of the indices which may be desired in many applications. Detailed derivations are provided as supplemental material (see S1 Methods).

**Differential gene-expression analysis.** To identify those public clonotypes which expand after SRBC application we performed a differential gene expression analysis using the R-

package *edgeR* [20]. Since the mathematical procedure is described in detail in [39] we give only a brief summary here. The algorithm is based on the assumption that for each clonotype, the distribution of the CNs follows a negative binomial distribution. For each clonotype, the null hypothesis that this distribution is not affected by the experimental conditions (SRBC-application, time point of immunization), was tested using a likelihood-ration test. To reduce the effect of unspecific proliferation phenomena or PCR artefacts, the threshold of significance for the likelihood-ratio test was determined as 0.005 [13]. In this analysis only those clonotypes were included which were detected in at least 75% of immunized animals. This way, we identified 40 clonotypes with significantly increased CN after immunization compared to control animals. In the original study, these criteria were fulfilled by 44 clonotypes [12], with the deviation caused by differences in data preprocessing including downsampling of the 7d group data. However, relative distributions remained comparable to the previous study: out of the 40 clonotypes an average of ~35 were detected in control animals and ~38 in immunized animals, with no significant differences between the three time points (not shown). While the mean CN per clonotype was considerably elevated at 3d and 4d to approximately 4- and 6-fold compared to control animals ($p < 4 \cdot 10^{-6}$ and $p < 6 \cdot 10^{-8}$), respectively, CNs and thus clonotype expression levels at 7d after immunization decreased to about 2-fold compared to the control level ($p < 4 \cdot 10^{-5}$).

## Supporting information

**S1 Fig. Logarithmic fractioning of the repertoire shows graduated shifting of certain parameters with increasing copy number and the intermediate part of the repertoire untouched by immunization-induced effects.**
(PDF)

**S2 Fig. Classification of total repertoires is successful only when weighted for copy numbers.**
(PDF)

**S1 Methods. Detailed derivations of the adapted statistic indices introduced in this study.** For statistic properties, which differ from those of the original versions, detailed analytic proofs are provided. Furthermore, we provide weighted versions (counting for copy numbers) for two of these indices.
(PDF)

**S1 Table. Number of extracted clonotypes and sequence reads of the 50 data sets after all preprocessing steps (including downsampling of the 7d group and removing clonotypes with CN 1).**
(PDF)

## Acknowledgments

We thank Kathrin Kalies and Johannes Textor for helpful comments on the manuscript and Silke Szymczak for revision of mathematical procedures.

## Author Contributions

**Conceptualization:** Martin Meinhardt, René Pagel, Jürgen Westermann.

**Data curation:** René Pagel.

**Formal analysis:** Martin Meinhardt.

**Methodology:** Martin Meinhardt.

**Project administration:** Cornelia Tune, Jürgen Westermann.

**Software:** Martin Meinhardt.

**Supervision:** Cornelia Tune, Jürgen Westermann.

**Visualization:** Martin Meinhardt.

**Writing – original draft:** Martin Meinhardt, Cornelia Tune.

**Writing – review & editing:** Lisa-Kristin Schierloh, Andrea Schampel, Jürgen Westermann.

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
