## [Decision Letter · Decision Letter 0]

9 Jun 2022

PONE-D-22-11828Tracking a diverse immune response against a complex cellular antigen in the murine T cell receptor repertoire: Over time, expanding private clones accumulate in the high but also the low copy number regionPLOS ONE

Dear Dr. Westermann,

Thank you for submitting your manuscript to PLOS ONE. After careful consideration, we feel that it has merit but does not fully meet PLOS ONE’s publication criteria as it currently stands. Therefore, we invite you to submit a revised version of the manuscript that addresses the points raised during the review process.

Some issues and questions arose with respect to methodology and interpretation of your work that require detailed changes and re-interpretation.

We look forward to receiving your revised manuscript.

Kind regards,

Jörg Hermann Fritz, Ph.D.

Academic Editor

PLOS ONE

Journal Requirements:

Additional Editor Comments:

Dear Dr. Westerman,

two experts in the field have reviewed your manuscript and consider the work highly interesting. However, a couple a issues and questions have been raised with respect to interpretation and clarity that need to be addressed and discussed.

with best regards

Jörg Fritz

Reviewers' comments:

Reviewer's Responses to Questions

**Comments to the Author**

1. Is the manuscript technically sound, and do the data support the conclusions?

Reviewer #1: Partly

Reviewer #2: Yes

2. Has the statistical analysis been performed appropriately and rigorously? 

Reviewer #1: Yes

Reviewer #2: Yes

3. Have the authors made all data underlying the findings in their manuscript fully available?

Reviewer #1: No

Reviewer #2: Yes

4. Is the manuscript presented in an intelligible fashion and written in standard English?

Reviewer #1: Yes

Reviewer #2: Yes

5. Review Comments to the Author

Reviewer #1: In this manuscript, Meinhardt et al., analyze the splenic TCR repertoire of mice post-immunization with sheep red blood cells. They adapt a set of statistical metrics with the aim of studying the dynamics of the SRBC-specific response over time and detect clonotypes contributing to this response. They also introduce a fractioning strategy allowing to identify the distribution of SRBC-related clonotypes within the repertoire. They posit that early after immunization, private antigen-specific clones accumulate in high copy numbers, whereas they are found in lower numbers at later timepoints.

There is high value in this paper in terms of highlighting the importance of looking beyond the top expanded clonotypes when studying an immune response over time, and the challenges that come with the search for response-specific clonotypes in a highly diverse TCR repertoire, which is like looking for a needle in a haystack.

However, I fear that the fractioning method that they put in place could only work for unsorted samples, as such repertoires represent a smooth and “conventional” distribution across all count fractions. In contrast, studying sorted populations of different sizes and phenotypes would imply differential clonotype count distributions and subsequently incomparable CN fraction sizes. Thus, using their strategy can only allow a global description of the repertoire modulation post-immunization, but do not allow a thorough investigation of a specific cell subset dominating the response.

I am curious about the dominant discriminatory contribution of the VJ usage in this study, as we would expect modifications of the CDR3 characteristics in response to an antigen rather than the VJ usage. Moreover, this contribution might be only reflecting the underlying differential contribution of the CD4/CD8 populations in the immune response and might not be the case when studying sorted populations.

The term “tracking” is used in the manuscript title. The way I see it, this would imply the tracking of specific clonotypes across time, which is not the case. Thus, this term, and the whole title which is bit too long, should maybe be reconsidered.

Overall, I think that the strategical aspects of this paper need to be looked at more carefully in order to determine the validity of the conclusions that are made and the hypotheses that are proposed.

Introduction:

“The specificity of the receptor of each single T cell is restricted to a small set of amino acid patterns”: Any references in support of this statement? To my knowledge, a T-cell can recognize up to 106 different pMHC complexes (Mason, 1998; Wooldridge et al., 2012). Moreover, the TCR can induce structural shifts allowing the recognition of different peptides as it is the case for the BM3.3 TCR (Archbold et al., 2009; Borbulevych et al., 2009, Reiser et al., 2003).

Methods:

1- Additional precisions on the MiXCR parameters used to align the datasets should be provided.

2- Was the down-sampling performed using the number of sequencing reads as a threshold and thus applied on the lists of reads pre-alignment? I think this is an important aspect to be clarified as in this case, the sampling threshold would take into account sequencing errors and unproductive sequences that might lead to an overestimation of the repertoire sizes.

Along with this point, the down-sampling applied on the repertoires 7d post-immunization could be causing the elimination of rare clonotypes that could otherwise be contributing to the immune response, particularly as significant perturbations were observed in the CNlow fraction at this timepoint.

3- “Each clonotype was assigned the V- and J- segments of the underlying nucleotide sequence of highest read count”. Why did the authors opt for this strategy? I think assigning the most expressed VJ combination to each “set of clonotypes” might overshadow other highly expressed combinations (was the VJ distribution looked at beforehand?) subsequently giving a higher importance to certain combinations in the RHIVJ analysis.

4- If I understood well, the repertoire fractioning was applied without taking into account the VJ genes. While I totally agree with the rationale of focusing on the CDR3 region while searching for relevant CDR3s in the context of an immunization in view of its center role in antigen recognition, taking into account the V-J usage in the identification of repertoire fractions could be important particularly as it is one of the two criteria on which this study is based.

5- It is mentioned that the number of clonotypes ranged from 50k to 110k per sample, however these numbers do not match with the ones plotted in Fig1A. A summary table of the number of sequences/clonotypes would be much clearer.

Figures:

Fig. 1:

1- Title: The use of the term “SRBC-specific” is not appropriate as no functional assays were performed to validate the specificity of the identified clones.

2- Fig1B: an “e” is missing in “sequence”

3- Was the arbitrary fractioning completely random or based on a certain percentage of the top clonotypes? I am curious about the number of clones being that homogeneous between experimental groups for the CNlow and CNmed but not the CNhigh fractions.

4- What does “clonotype” refer to in this section? If I refer to the materials and methods, a clonotype is a “set of T cells with identical CDRβ region”, and thus does not include the V and J genes. With that being said, “two clonotypes are defined as similar if the LvD of their CDR regions is at maximum 1”, how could two clonotypes be completely similar and thus have a LvD=0 if the VJ genes are not accounted? Shouldn’t they be considered as one clonotype in this case? More precision on the definition of a clonotype is needed in this section to clarify these subtilties.

5- I would not consider that the RHIVJ can capture” sequence similarity”, but rather the diversity in the gene usage. Two clonotypes expressing the same VJ combination but having completely different CDR3 sequences cannot be considered as similar, particularly when the specificity is mostly encoded by the CDR3 sequence.

6- A visualization of the LvD clusters as a complement to the RHILD could bring additional information on the size/density of the formed clusters and the effect of the sequence similarity reduction observed 3d post-immunization compared to the PBS group on the cluster’s architecture. Considering that private clones displace public ones (as stated in the discussion), big dense clusters should be fractioned into multiple small clusters.

7- The use of the term “homogenization” in reference to a decrease in the number of nucleotide CDR3 sequences encoding for the same amino acid sequence is confusing, as by homogenous one would think of a homogeneous number of nucleotide sequences across all amino acid clonotypes in the repertoire. It is however not the case herein.

Fig 2:

1- It would be easier to follow through the results if the main text and the figures have the same CN nomenclature (either use 1-9 or 2-512).

2- It would be interesting to look at the Jaccard scores between the CNhigh fraction at day 3 and CNlow at day 7 to track whether the potentially antigen-related expansions are the ones that are detected at low frequency at a later timepoint post-immunization.

3- I’m curious about the differences observed in the 64<cn<512 absence="" all="" and="" any="" clones="" compared="" control="" differences="" for="" fractions="" group.="" in="" indices="" number="" of="" other="" significant="" terms="" the="" these="" three="" to="" within="">Moreover, as RHIVJ shows significant differences for fractions with CN>64, I would assume that the perturbation in the VJ usage is not directly linked or caused by the potential SRBC-related clonotypes that are expanded 3-4 days post-immunization and which are observed within the CN>256 fraction. This might be caused by the V-J assigning strategy.

4- The superior discriminatory power of RHIVJ compared to RHILD and CDI can be a reflection of the differential contribution of the CD8 and CD4 populations in the immune response against SRBC as the study was done on unsorted T cells. Has the CD8/CD4 ratio been looked at?

Fig 3:

1- Precisions on the methods used for both unsupervised and supervised classification within the main text are needed to further clarify what was done in this section.

2- The choice of the “relevant” fractions is arbitrary in my sense and could be the reason behind the discordant results obtained on both fractions when looking at the VJ usage and nucleotide coding. In fig2 B, significant p values are obtained with CDI on the CN< 4 and CN>256 fractions whereas significant differences are observed with RHIVJ on CN<16 and CN>64. This could explain why VJ usage but not nucleotide coding performs well on the chosen Xtop fraction (CN>64), whereas nucleotide coding classifies well on the Xbottom (CN<4) fraction. Thus, I think the choice of fractions should take into account the results obtained for each index in fig2.

3- For the Xbottom fraction, 3d and 4d post-immunization repertoires could be used as a control in the supervised classification analysis by comparing the repertoire of each timepoint to the control group. Based on the results in fig2B, values should be similar to the ones obtained when applying random labeling.

4- FigS2: Did the authors perform the classification analysis using the nucleotide coding criteria on the total repertoire?

Fig4:

1- A brief description of the identification strategy of the “public SRBC-specific” clonotypes is needed in the results section. Was the strategy timepoint-dependent, i.e. applied on each timepoint compared to the control group?

2- Does this strategy take into account the CN of the clonotypes across mice within the same experimental group? For example, a clonotype identified as significantly more present in the immunized group but in totally different CN fractions across the mice (CNlow and CNhigh) might not have the same weight nor contribution in the SRBC response than other clonotypes that are found within the CNhigh fraction in all mice.

3- In view of the low Jaccard scores shown in fig1, I find it surprising that only 40 clonotypes were found to be enriched in the immunized group. It would maybe be interesting (if it is not already the case) to look at each time point independently as their behavior is not similar across all previous analyses.

4- Again, the term “specific public clones” is not appropriate as the specificity was not tested by functional assays.

5- I do not think that the observations in fig 2 and 3 reflect the behavior of “private clonotypes” exclusively as no prior filtering was applied to select such clonotypes as it is the case in fig 4 for the public ones. A more appropriate comparison could be performed between the identified 40 public clones and a list of CDR3s with no significant enrichment in the immunized group in order to confirm the accumulation of the public clones within the highest fraction.

6- The inclusion of the RHIVJ results from fig2 is misleading and seems unnecessary in this analysis, particularly as the number of clonotypes and the RHIVJ did not show concordant observations in fig 2. It would be more interesting to look at the RHILD of the 40 public clonotypes compared to a list of non-enriched CDR3s. Furthermore, the search for amino acid motifs that are shared by the private clonotypes could reveal their potential implication in the SRBC response, or on the contrary reveal their rather bystander activation.</cn<512>

Reviewer #2: Summary

In this manuscript, Meinhardt et al. reported a novel approach to estimate the diversity of TCRs using CDR3β and VJ usage. The authors utilize this model to reveal the T cell repertoire dynamics post-immunization. This approach can potentially be also used in studying a variety of immune responses. The manuscript is suitable for publishing in PLOS ONE.

Please see attachment for comments.

6. PLOS authors have the option to publish the peer review history of their article (what does this mean?). If published, this will include your full peer review and any attached files.

Reviewer #1: No

Reviewer #2: No

---

## [Decision Letter · Decision Letter 1]

5 Aug 2022

The splenic T cell receptor repertoire during an immune response against a complex antigen: Expanding private clones accumulate in the high and low copy number region

PONE-D-22-11828R1

Dear Dr. Westermann,

We’re pleased to inform you that your manuscript has been judged scientifically suitable for publication and will be formally accepted for publication once it meets all outstanding technical requirements.

Kind regards,

Jörg Hermann Fritz, Ph.D.

Academic Editor

PLOS ONE

Additional Editor Comments (optional):

Reviewers' comments:

Reviewer's Responses to Questions

**Comments to the Author**

1. If the authors have adequately addressed your comments raised in a previous round of review and you feel that this manuscript is now acceptable for publication, you may indicate that here to bypass the “Comments to the Author” section, enter your conflict of interest statement in the “Confidential to Editor” section, and submit your "Accept" recommendation.

Reviewer #1: All comments have been addressed

Reviewer #2: All comments have been addressed

2. Is the manuscript technically sound, and do the data support the conclusions?

Reviewer #1: (No Response)

Reviewer #2: Yes

3. Has the statistical analysis been performed appropriately and rigorously? 

Reviewer #1: (No Response)

Reviewer #2: Yes

4. Have the authors made all data underlying the findings in their manuscript fully available?

Reviewer #1: (No Response)

Reviewer #2: Yes

5. Is the manuscript presented in an intelligible fashion and written in standard English?

Reviewer #1: (No Response)

Reviewer #2: Yes

6. Review Comments to the Author

Reviewer #1: All comments have been addressed. Two minor comments to be taken into account:

-Title: if clones are expanded then they shouldn't be in the low copy number fraction... Maybe review this term in the title.

-Fig1: specify the logarithmic scale in fig1A

Reviewer #2: Summary

In this manuscript, Meinhardt et al. utilized a new model to estimate the dynamic diversity of TCRs post-immunization. The authors have addressed some of the concerns and comments in my previous review. The revised version has improved clarity and more detailed methodology. The manuscript is suitable for publishing on PLOS ONE.

Comments for the authors (Revision is not needed):

I am satisfied with the authors’ responses. No further edit is needed.

7. PLOS authors have the option to publish the peer review history of their article (what does this mean?). If published, this will include your full peer review and any attached files.

Reviewer #1: No

Reviewer #2: No

---

## [Editor Report · Acceptance letter]

12 Aug 2022

PONE-D-22-11828R1 

The splenic T cell receptor repertoire during an immune response against a complex antigen: Expanding private clones accumulate in the high and low copy number region 

Dear Dr. Westermann:

I'm pleased to inform you that your manuscript has been deemed suitable for publication in PLOS ONE. Congratulations! Your manuscript is now with our production department. 

Kind regards, 

on behalf of

Dr. Jörg Hermann Fritz 

Academic Editor

PLOS ONE